# Perceived Qualities, Visitation and Felt Benefits of Preferred Nature Spaces during the COVID-19 Pandemic in Australia: A Nationally-Representative Cross-Sectional Study of 2940 Adults

**Xiaoqi Feng** [1,2,3,*] **and Thomas Astell-Burt** [2,3]

1. School of Population Health, University of New South Wales, Sydney, NSW 2052, Australia
2. Population Wellbeing and Environment Research Lab (PowerLab), NSW, Australia; thomasab@uow.edu.au
3. School of Health and Society, University of Wollongong, Wollongong, NSW 2522, Australia
* Correspondence: xiaoqi.feng@unsw.edu.au

**Abstract:** We investigated how the perceived quality of natural spaces influenced levels of visitation and felt benefits during the COVID-19 pandemic in Australia via a nationally representative online and telephone survey conducted on 12–26 October (Social Research Centre's Life in Australia[TM] panel aged > 18 years, 78.8% response, *n* = 3043). Our sample was restricted to those with complete information (*n* = 2940). Likert scale responses to 18 statements regarding the quality of local natural spaces that participants preferred to visit were classified into eight quality domains: access; aesthetics; amenities; facilities; incivilities; potential usage; safety; and social. These domains were then summed into an overall nature quality score (mean = 5.8, range = 0–16). Associations between these quality variables and a range of nature visitation and felt benefits were tested using weighted multilevel models, adjusted for demographic and socioeconomic confounders. Compared with participants in the lowest perceived nature quality quintile, those in the highest quality quintile had higher odds of spending at least 2 h in their preferred local nature space in the past week (Odds Ratio [OR] = 3.40; 95% Confidence Interval [95%CI] = 2.38–4.86), of visiting their preferred nature space almost every day in the past four weeks (OR = 3.90; 2.77–5.47), and of reporting increased levels of nature visitation in comparison with before the COVID-19 pandemic (OR = 3.90; 2.54–6.00). Participants in the highest versus lowest perceived nature quality quintile also reported higher odds of feeling their visits to nature enabled them to take solace and respite during the pandemic (OR = 9.49; 6.73–13.39), to keep connected with their communities (OR = 5.30; 3.46–8.11), and to exercise more often than they did before the pandemic (OR = 3.88; 2.57–5.86). Further analyses of each quality domain indicated time in and frequency of visiting nature spaces were most affected by potential usage and safety (time in nature was also influenced by the level of amenity). Feelings of connection and solace were most affected by potential usage and social domains. Exercise was most influenced by potential usage, social and access domains. In conclusion, evidence reported in this study indicates that visits to nature and various health-related benefits associated with it during the COVID-19 pandemic were highly contingent upon numerous qualities of green and blue spaces.

**Keywords:** nature; green space; blue space; quality; visit; benefit; solace; physical activity; social connection; safety



## 1. Introduction

Epidemiological, experimental and ethnographic research accumulated over many decades indicates that contact with nature (i.e., green and blue spaces) can provide relief from stress and restore depleted cognitive resources, enrich lives with more socially and physically active recreational pursuits, and endow communities with cleaner and cooler microclimates [1–3]. The net-health benefits of these entwined pathways include lower risks of cardiovascular diseases [4–11], diabetes [12–14] and death [15], while a smaller

number of studies also identify lower risks of loneliness [16,17], cognitive decline and dementia [18–27].

Numerous studies around the world have shone a light on how contact with nature has played an important role in coping with protracted socioeconomic upheaval and significant emotional distress during the COVID-19 pandemic [28]. Many have zeroed in on the mental health benefits of nature during this difficult period [29]. Some work has focused on the inequalities that were exacerbated by lockdowns [30] and the compensation of private gardens for a lack of access to public green spaces [31,32]. Others have shown how the rapid transition to 'remote work' in some countries enabled more people to benefit from spending time in nature [33]. Notable for its absence from current research, however, was the critical issue of green and blue space *quality*, which was not mentioned once in a comprehensive narrative review of studies published [28]. To our knowledge, only one study has examined the role of green space quality during the pandemic [29].

Green and blue space quality is a critical issue for health and mental health in particular, as many previous studies conducted before the COVID-19 pandemic demonstrate [34–37]. Qualitative evidence bears this out in ways that are largely absent from the epidemiological literature [38,39]. For example, many people will visit natural settings that they identify as providing sources of non-judgmental, ego-free and dependable support [40]; it would be erroneous to consider all green and blue spaces as affording such precious experiences. Meanwhile, some natural settings can, as a result of regular visitation over time and over generations, become invested with individual and shared meanings that (re)generate feelings of connection and belonging within and between nearby communities [41]. Clearly, not all green and blue spaces share equal value. If a nearby green or blue space has a deficit of what people might need or desire (i.e., 'good' qualities) and/or an abundance of things they might seek to avoid (i.e., 'bad' qualities), a situation may transpire where there are lots of nature spots nearby but this does not necessarily translate into better health.

Accordingly, we sought to look beyond the sole issues of availability and accessibility to investigate more holistically the degree to which the nature–health association during the COVID-19 pandemic was dependent upon the various qualities of the nearby natural settings that people prefer to visit. We examined the extent to which different aspects of visitation and contrasting health benefits were contingent upon the presence of a range of specific qualities, both good and bad. We hypothesized that people tended to visit and benefit more from green and/or blue spaces that they felt had more of the qualities they value.

## 2. Materials and Methods

### 2.1. Data

A nationally-representative survey of the Australian adult population aged ≥ 18 years was conducted between 12 and 26 October 2020 via the Social Research Centre's Life in Australia[TM] panel. Participation was in the English language only. Panel members were recruited in 2016 via their landline or cell phone using a dual-frame random digit dialling design (RDD) with a 30:70 split between landline and cell phone sample frames. Respondents in households were selected using an alternating next or last birthday approach via the landline method wherein households were occupied by at least two residents in scope. The phone answerer was the selected respondent via the cell phone method. In each case, invitation to join the panel was for one member per household only. The panel was refreshed in 2018 using mobile phone RDD only and again in 2019 with online-only participants using a G-NAF (Geocoded National Address File) sample frame and push-to-web methodology. In each of these cases, refreshment was required to balance demographics of the panel with respect to the Australian population. Online panel members were invited to participate in the survey via email and SMS, followed by emails, telephone calls, voicemails and SMS in week 2 of the survey period to encourage completion. The Social Research Centre's interviewers and supervisors had received training in the Life in Australia[TM] panel, survey procedures and sample management protocols, respondent liaison procedures, strategies

to maintain cooperation, and detailed examination of the survey questionnaire developed by the researchers. An incentive of a supermarket or department store gift card, direct payment into a PayPal account, or donation to a designated charity was offered to all panel members to the value of AUD 10.00 each. Ethical approval for the survey was granted by the University of Wollongong HREC.

Approximately 78.8% (*n* = 3043) of the Life in Australia$^{TM}$ panel participated in our survey (95.0% completing online, 5.0% completing via telephone). A total of 19.9% of panel members could not be contacted during the survey period and 1.3% of invited members declined to participate. Response propensity weights were constructed by the Social Research Centre using logistic regression to limit the impact of non-participation on sample representativeness, taking into account geography, age group, gender, annual household income, citizenship status, language(s) spoken other than English, country of birth, Aboriginal or Torres Strait Islander status, number of adults and children in the household, employment status, marital status, highest education, television viewing and internet browsing habits, smoking and drinking status, general health, life satisfaction, early adopter status, caregiving, disability status, volunteer status, concession card status, and telephony status. Our sample focussed on people with full outcome data (*n* = 2940; 96.6%).

### 2.2. Outcomes: Nature Space Visitation and Felt Benefits

Six binary outcomes describing participants' visitation of, and felt benefits from nearby natural settings during the COVID-19 pandemic were examined. Given evidence indicating at least 120 min of time in nature may support general health, we used this as a cut-point for responses to the question: "*Approximately how many hours did you spend in green spaces and/or blue spaces in total over the last 7 days?*" We allowed for the possibility that for some people, the natural setting(s) they would prefer to visit might not be accessible for multiple reasons and so responses to the following question were also examined: "*In the past four weeks (including the weekends), how often have you visited your preferred local green space and/or blue space?*" Responses were classified as 'at least once a week' (combining 'almost daily' and '1–4 times weekly') or 'less than once a week' (combining '2–3 times in the past month', 'once or less in the past month', and 'never'). The third visitation-focussed outcome was used to determine if participants had increased their visit frequency since the pandemic began with the following question: "*Since the COVID-19 pandemic and social distancing began in Australia, to what extent, if at all, do you agree or disagree with each of the following statements? (A) I now visit green spaces and/or blue spaces more often than before the COVID-19 pandemic*". This question set also included three statements pertaining to felt benefits, as follows: "*(B) Green spaces and/or blue spaces have helped me to stay connected with my neighbours during the COVID-19 pandemic. (C) Green spaces and/or blue spaces have brought me solace and respite in these challenging times. (D) I now walk and/or exercise in green spaces and/or blue spaces more frequently than before the COVID-19 pandemic*".

### 2.3. Qualities of Nearby Natural Spaces

Eighteen different quality indicators on the nearby natural setting participants preferred to visit were measured in the survey (Table 1). These indicators were used to construct eight different quality domains adopted from a published green space quality auditing tool [42]. The survey question was "*Thinking of the green space and/or blue space you prefer to visit most often and your experiences in it, how much do you agree or disagree with the following statements?*" The first domain was "access", for which participants were asked whether their preferred nature spaces were well-connected by public transport, footpaths and road crossing points. The second domain, "aesthetics", examined views concerning potential for exploration and interesting discoveries, as well as pleasant natural vistas and biodiverse soundscapes. "Amenities" were the third domain to permit acknowledgement of adjacent reasons people might visit the natural setting for, such as before or after shopping and dining out. The amenities domain also included a separate question on provision of shade along footpaths from tree canopy cover. The fourth domain described "facilities",

including those used for physical activity, public bathrooms, and seating. "Incivilities" were the fifth domain and were measured by a single item pertaining to perception of quality and maintenance of the natural space. The sixth domain described various types of "usage" including breaks from day-to-day routines, opportunities to feel some distance from cognitive demands, and spaces that children can play outdoors and/or that a participant feels they can walk and/or exercise in. The seventh domain attended to "safety", with specific focus on safety during the evening/night. The eighth and final domain considered "social" factors, such as whether the green/blue space was viewed as a shared setting for neighbours, friends and/or family to meet. Where possible, these questions were derived from existing literature. For instance, "*there is much to explore and discover there*" in the aesthetics domain was drawn from Hartig's perceived restorativeness scale [43]. The answer set to all eighteen quality indicators was a five-point Likert scale.

**Table 1.** Eighteen nature space quality indicators nested within eight quality domains.

---

*"Thinking of the green space and/or blue space you prefer to visit most often and your experiences in it, how much do you agree or disagree with the following statements?"*
*[strongly disagree, disagree, neither agree nor disagree, agree, strongly agree]*

*Domain 1: Access*
"It is well connected by footpaths and safe road crossing points"
"Public transport is available nearby"

*Domain 2: Aesthetics*
"There is much to explore and discover there"
"My attention is drawn to many interesting things there"
"It is a place I can enjoy watching and/or listening to wildlife (e.g., birds)"

*Domain 3: Amenities*
"There are cafes, and/or shops, and/or supermarkets and/or restaurants nearby"
"There is lots of tree canopy along footpaths that provide shade from heat and direct sunlight"

*Domain 4: Facilities*
"There are free or low cost recreation facilities, such as outdoor gyms, sports grounds and/or swimming pools in it or nearby"
"There are public toilets available in it or nearby"
"There are benches in it or nearby on which I can sit and relax"

*Domain 5: Incivilities*
"I consider it to be high quality and well maintained"

*Domain 6: Potential usage*
"Spending time there gives me a break from my day-to-day routine"
"This is a place to get away from the things that usually demand my attention"
"It is a good place for children to play outdoors"
"I go there for walks and/or to exercise"

*Domain 7: Safety*
"This is a place I feel safe to visit during the evening/night"

*Domain 8: Social*
"It is a social hub for the local community"
"This is a place to spend time with friends and/or family"

---

We classified each indicator to "disagree/ambivalent" (scoring zero, combining "strongly disagree", "disagree", and "neither agree nor disagree"), "agree" (scoring 1) and "strongly agree" (scoring 2). Participants' mean scores across all quality indicators within each domain were calculated. Degree of correlation between domain means was assessed using Pearson's correlation coefficients (Table 2), from which it was evident that most domains were weak-

to-moderately correlated (e.g., the only correlation >0.6 was for the aesthetics and usage domains: coefficient = 0.619, *p*-value < 0.001). Those domain means were then summed across all domains to give a total quality score for nearby green and blue spaces. This total quality score was normally distributed with an overall mean of 5.84 (standard deviation = 3.05) and ranged from zero to 16. We classified it into quintiles, for which the interval means and other parameters are reported in Table 3.

**Table 2.** Pearson's correlation coefficients for quality domains of natural settings.

|  | Access | Aesthetics | Amenities | Facilities | Incivilities | Potential Usage | Safety | Social |
|---|---|---|---|---|---|---|---|---|
| Access | 1 | | | | | | | |
| Aesthetics | 0.164 | 1.000 | | | | | | |
| *p*-value | <0.001 | | | | | | | |
| Amenities | 0.462 | 0.354 | 1.000 | | | | | |
| *p*-value | <0.001 | <0.001 | | | | | | |
| Facilities | 0.520 | 0.282 | 0.507 | 1.000 | | | | |
| *p*-value | <0.001 | <0.001 | <0.001 | | | | | |
| Incivilities | 0.425 | 0.338 | 0.427 | 0.520 | 1.000 | | | |
| *p*-value | <0.001 | <0.001 | <0.001 | <0.001 | | | | |
| Potential Usage | 0.375 | 0.619 | 0.441 | 0.424 | 0.483 | 1.000 | | |
| *p*-value | <0.001 | <0.001 | <0.001 | <0.001 | <0.001 | | | |
| Safety | 0.124 | 0.257 | 0.147 | 0.162 | 0.215 | 0.273 | 1.000 | |
| *p*-value | <0.001 | <0.001 | <0.001 | <0.001 | <0.001 | <0.001 | | |
| Social | 0.396 | 0.396 | 0.450 | 0.559 | 0.477 | 0.494 | 0.216 | 1 |
| *p*-value | <0.001 | <0.001 | <0.001 | <0.001 | <0.001 | <0.001 | <0.001 | |

**Table 3.** Description of the nature space total quality score quintile intervals.

| Quintiles | Mean | Standard Deviation | Quintile Bounds | |
|---|---|---|---|---|
| 1 (low) | 2.02 | 1.00 | 0.00 | 3.30 |
| 2 | 4.11 | 0.45 | 3.33 | 4.83 |
| 3 | 5.53 | 0.39 | 4.87 | 6.17 |
| 4 | 7.08 | 0.57 | 6.20 | 8.03 |
| 5 (high) | 10.53 | 1.83 | 8.07 | 16.00 |

### 2.4. Confounders

Variables that denote factors known to influence both human behaviour and psychological and social wellbeing, and also where people live and access to green and/or blue space, were measured using survey responses. These included gender, age, country of birth, language spoken at home, relationship status, highest educational qualification, annual household income, economic status (e.g., employed, retired, unemployed), perceived financial difficulties, housing type (e.g., house, flat), and urban/rural. The urban/rural variable was extended to 15 categories to account for substantial geographical variations across the states and territories of Australia, differentiating between participants living in major cities (e.g., Sydney, Melbourne) from those living in regional and rural areas of the same states (e.g., Rest of New South Wales, Rest of Victoria).

### 2.5. Statistical Analysis

Cross-tabulations, percentages and means were used to describe the study sample and the patterning of the total nature space quality scores across participants' characteristics. Weighted linear regressions were used to assess associations between the total nature space quality scores and participants' characteristics. Separate weighted logistic regressions were then used to examine associations between each of the nature space visitation and felt benefit outcomes with the quality scores, adjusting for confounding variables. All analyses were conducted in Stata V.14 (StataCorp., College Station, TX, USA).

## 3. Results

### 3.1. Sample Description and Differences in Nature Space Total Quality Scores

Weighted descriptive statistics of the study sample are reported in Table 4, as well as unadjusted mean nature space total quality scores and adjusted coefficients from a weighted multiple linear regression. The ratio of females to males is almost equal. About 53% of the sample was aged between 25 years and 54 years. About two-thirds of the sample were born in Australia and nearly four-fifths did not speak a language other than English at home. Approximately 71% of the sample were in a couple with or without children, whereas just over 15% were living in single-person households. University degrees were held by 27.6% of participants whereas the highest qualification for 13.2% was fewer than 12 years of education. In total, 59% of the sample had annual household incomes up to AUD 100 k, whereas 34.6% had incomes greater than or equal to AUD 101 k per year. Unemployment was at 9.5%, retirement at 20.7%, employment at 61.1% and those living with disability at 2.6%. Employment varied with respect to remote work, with 29.7% having no remote work option whereas 13.9% working remotely full-time. Nearly 10% of the sample reported financial difficulty relative to 26.5% who were comfortable. Approximately 37.6% of the sample was resident in the cities of Sydney or Melbourne, with 66.7% living in major cities.

**Table 4.** Sample description, mean quality of preferred nearby natural setting, and adjusted differences, weighted for national representativeness.

| Total Sample *n* = 2940 | *n* (%) | Mean (SE) | Coef (95%CI) [*p*-Value] |
|---|---|---|---|
| Gender (ref: Female) | 49.4% | 5.6 (0.1) | |
| Male | 50.5% | 6.0 (0.1) | 0.432 (0.138, 0.726) [0.004] |
| Other | 0.2% | 6.9 (1.2) | 0.950 (−1.091, 2.990) [0.361] |
| Age group (ref: 18–24 years) | 10.0% | 5.2 (0.3) | |
| 25–34 years | 18.5% | 6.2 (0.2) | 0.977 (0.225, 1.729) [0.011] |
| 35–44 years | 18.1% | 6.2 (0.2) | 0.834 (0.105, 1.563) [0.025] |
| 45–54 years | 16.3% | 5.9 (0.2) | 0.764 (0.036, 1.491) [0.040] |
| 55–64 years | 15.1% | 5.5 (0.1) | 0.515 (−0.192, 1.222) [0.153] |
| 65–74 years | 14.0% | 5.4 (0.2) | 0.604 (−0.231, 1.440) [0.156] |
| ≥75 years | 6.9% | 5.4 (0.2) | 0.588 (−0.321, 1.496) [0.205] |
| Undetermined | 1.1% | 6.9 (0.9) | 1.792 (−0.046, 3.630) [0.056] |
| Country of birth (ref: Australia) | 66.1% | 5.7 (0.1) | |
| Overseas, not English-speaking | 19.1% | 5.8 (0.2) | −0.392 (−0.946, 0.162) [0.165] |
| Overseas, English-speaking | 14.7% | 5.9 (0.2) | 0.161 (−0.296, 0.618) [0.489] |
| Undetermined | 0.2% | 7.6 (1.5) | 0.638 (−2.987, 4.263) [0.730] |
| Language other than English at home (ref: Yes) | 20.1% | 6.1 (0.2) | |
| No | 79.9% | 5.7 (0.1) | −0.375 (−0.946, 0.196) [0.198] |
| Relationship status (ref: Living alone) | 15.2% | 5.5 (0.2) | |
| Alone with kids | 6.9% | 5.6 (0.3) | 0.232 (−0.426, 0.890) [0.489] |
| Couple without kids | 27.7% | 5.6 (0.1) | 0.139 (−0.287, 0.565) [0.521] |
| Couple with kids | 43.4% | 6.0 (0.1) | 0.413 (−0.037, 0.864) [0.072] |
| Cohabiting, unrelated | 2.7% | 6.3 (0.4) | 0.592 (−0.212, 1.397) [0.149] |
| Other | 4.1% | 4.9 (0.4) | −0.576 (−1.380, 0.228) [0.160] |
| Highest educational qualification (ref: <Year 12) | 13.2% | 5.5 (0.2) | |
| Year 12 | 19.0% | 5.6 (0.2) | −0.020 (−0.574, 0.533) [0.943] |
| Advanced diploma/certificate | 37.2% | 5.7 (0.1) | 0.069 (−0.385, 0.522) [0.767] |
| Bachelor degree | 18.8% | 6.2 (0.1) | 0.334 (−0.182, 0.850) [0.204] |
| Postgraduate degree | 8.8% | 6.1 (0.1) | 0.195 (−0.360, 0.751) [0.491] |
| Undetermined | 3.0% | 5.0 (0.3) | −0.310 (−1.008, 0.388) [0.384] |

**Table 4.** *Cont.*

| Total Sample *n* = 2940 | *n* (%) | Mean (SE) | Coef (95%CI) [*p*-Value] |
|---|---|---|---|
| Annual household income (ref: ≤50 K) | 27.1% | 5.5 (0.1) | |
|    AUD 51 K–AUD 100 K | 31.9% | 5.7 (0.1) | −0.135 (−0.530, 0.259) [0.501] |
|    AUD 101 K–AUD 150 K | 18.0% | 6.1 (0.2) | −0.115 (−0.627, 0.396) [0.658] |
|    ≥AUD 151 K | 16.6% | 6.1 (0.2) | −0.136 (−0.776, 0.503) [0.675] |
|    Undetermined | 6.4% | 5.4 (0.3) | −0.337 (−0.996, 0.322) [0.316] |
| Economic status (ref: Employed, never remotely) | 29.7% | 5.8 (0.2) | |
|    Employed, work remotely sometimes | 11.0% | 6.0 (0.2) | 0.091 (−0.469, 0.650) [0.751] |
|    Employed, work remotely often | 6.5% | 6.2 (0.3) | 0.228 (−0.386, 0.842) [0.466] |
|    Employed, work remotely always | 13.9% | 6.3 (0.2) | 0.210 (−0.281, 0.701) [0.402] |
|    Unemployed | 9.5% | 5.3 (0.3) | −0.377 (−0.993, 0.239) [0.231] |
|    Retired | 20.7% | 5.4 (0.1) | −0.183 (−0.758, 0.393) [0.534] |
|    Disabled | 2.6% | 4.9 (0.3) | −0.636 (−1.325, 0.053) [0.070] |
|    Other | 5.3% | 5.9 (0.3) | 0.121 (−0.505, 0.747) [0.705] |
|    Undetermined | 0.9% | 6.6 (0.7) | 0.812 (−0.615, 2.238) [0.265] |
| Economic difficulty (ref: Comfortable) | 26.5% | 6.2 (0.1) | |
|    Doing ok | 44.2% | 5.7 (0.1) | −0.460 (−0.809, −0.111) [0.010] |
|    Getting by | 19.3% | 5.6 (0.2) | −0.560 (−0.990, −0.131) [0.011] |
|    Difficult | 9.9% | 5.3 (0.2) | −0.801 (−1.398, −0.205) [0.009] |
|    Undetermined | 0.2% | 3.8 (0.5) | −2.308 (−3.715, −0.902) [0.001] |
| Housing (ref: House) | 75.0% | 5.8 (0.1) | |
|    Flat | 17.7% | 6.0 (0.2) | 0.335 (−0.089, 0.760) [0.122] |
|    Farmhouse | 5.2% | 4.8 (0.3) | −0.936 (−1.470, −0.403) [0.001] |
|    Retirement village | 0.9% | 5.4 (0.5) | −0.202 (−1.300, 0.895) [0.718] |
|    Other | 1.2% | 5.3 (0.7) | −0.044 (−1.305, 1.217) [0.945] |
| Geographic area (ref: Greater Sydney) | 18.8% | 5.8 (0.2) | |
|    Rest of New South Wales | 13.2% | 5.4 (0.2) | −0.072 (−0.603, 0.459) [0.790] |
|    Greater Melbourne | 18.8% | 6.2 (0.2) | 0.352 (−0.141, 0.845) [0.162] |
|    Rest of Victoria | 7.4% | 5.5 (0.3) | 0.006 (−0.610, 0.623) [0.984] |
|    Greater Brisbane | 11.1% | 5.6 (0.2) | −0.106 (−0.699, 0.487) [0.726] |
|    Rest of Queensland | 8.9% | 5.8 (0.2) | 0.182 (−0.387, 0.752) [0.530] |
|    Greater Adelaide | 5.7% | 5.6 (0.3) | −0.016 (−0.649, 0.617) [0.960] |
|    Rest of South Australia | 1.3% | 6.1 (0.5) | 0.508 (−0.588, 1.604) [0.363] |
|    Greater Perth | 10.3% | 6.0 (0.2) | 0.173 (−0.407, 0.752) [0.559] |
|    Rest of Western Australia | 1.5% | 5.1 (0.5) | −0.319 (−1.362, 0.724) [0.549] |
|    Greater Hobart | 0.7% | 6.6 (0.6) | 1.027 (−0.191, 2.245) [0.098] |
|    Rest of Tasmania | 1.0% | 5.8 (0.5) | 0.372 (−0.632, 1.377) [0.467] |
|    Greater Darwin | 0.2% | 5.0 (1.0) | −0.763 (−2.610, 1.083) [0.418] |
|    Rest of Northern Territory | 0.0% | 5.4 (1.1) | −0.533 (−2.139, 1.072) [0.515] |
|    Australian Capital Territory | 1.1% | 5.3 (0.4) | −0.140 (−1.060, 0.779) [0.765] |
| Constant | | | 5.330 (4.178, 6.482) [<0.001] |

SE: Standard Error; 95%CI: 95% Confidence Interval; Note: all means, standard errors, regression coefficients and 95% confidence intervals are weighted for national representativeness.

The mean nature quality scores tended to be higher among males in comparison with females (Table 4). Mean nature quality scores also tended to be higher among participants

aged 25–54 years in comparison to those aged 18–24 years, participants who felt their financial circumstances were comfortable relative to those who were not, and those in houses or flats relative to a farmhouse. Variations in mean nature space quality scores between other demographic and socioeconomic groups were small and not statistically significant.

### 3.2. Associations between Nature Space Quality Scores and Visitation and Felt Benefits

Higher quintiles of nature space total quality scores were consistently and positively associated with each of the visitation and felt benefit outcome variables, after adjustment for confounding variables and weighted for national representativeness (Figure 1).

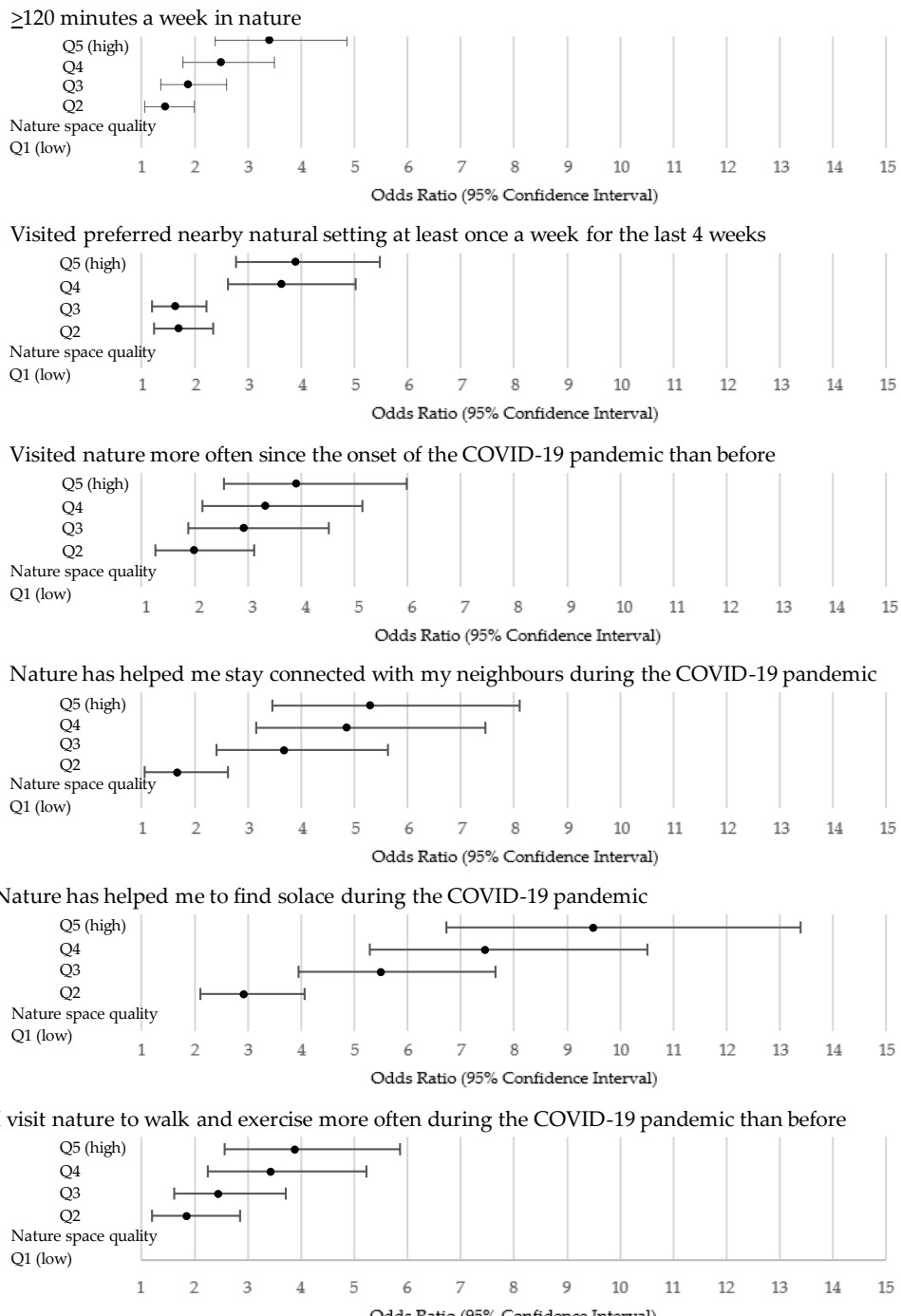

**Figure 1.** Adjusted associations between nature space total quality scores and visitation and felt benefits, weighted for national representativeness.

In comparison with quintile 1 (i.e., lowest quality), participants in quintile 5 (i.e., highest quality) had 3.4 times greater odds of spending two hours or more in nature a week (odds ratio [OR] 3.40, 95% confidence interval [95%CI] 2.38–4.86), 3.9 times greater odds of visiting their nearby preferred nature setting at least once a week for the last four weeks (OR 3.90, 95%CI 2.77–5.47), and 3.9 times greater odds of reporting that they visit green and blue spaces more often now than before the COVID-19 pandemic (OR 3.90, 95%CI 2.54–5.99). In order of magnitude, participants in quintile 5 compared with quintile 1 for nature space total quality scores had 9.5 times greater odds of reporting that time in nature had been a source of solace during the pandemic (OR 9.49, 95%CI 6.73–13.39; wider confidence intervals are in part indicative of smaller sample sizes), 5.3 times greater odds of reporting that green and blue spaces had enabled them to stay connected with their neighbors (OR 5.30, 95%CI 3.46–8.11), and 3.9 times greater odds of reporting more engagement in physical activity now than before the pandemic (OR 3.88, 95%CI 2.57–5.86).

Associations between each nature space quality domain and the six visitation and felt benefit outcome variables are reported in Table 5. Models were initially fitted for each outcome and quality domain score separately, followed by models that adjusted for all quality domain scores simultaneously. Every quality domain score was associated with more favorable outcomes when analyzed in isolation and adjusted for confounding. Many domains were no longer statistically significantly associated with the outcomes after adjusting for other domains. Potential usage was the only domain that was consistently associated with increased odds of all outcomes. The magnitude of odds ratios for potential usage were also consistently larger than in other domains. The access domain was only associated with taking exercise in nature more often than before the COVID-19 pandemic. Aesthetics, facilities and incivilities were not associated with any of the outcomes, while amenities were only associated—negatively—with the odds of spending at least two hours a week in nature. Higher levels of safety were important for both achieving at least two hours a week in nature and for visiting the preferred nearby natural setting at least once a week for the last four weeks, but not for any of the other outcomes. The social domain was positively associated with staying connected, finding solace, and taking more physical activity in nature, but none of the visitation-focused outcomes.

**Table 5.** Adjusted associations between qualities of nearby natural settings and visitation and felt benefits, weighted for national representativeness.

| | Single Quality Domain Model | Multi Quality Domain Model |
| --- | --- | --- |
| Visitation and felt benefit | Odds Ratio (95% Confidence Interval) [*p*-value] | |
| ≥120 min a week in nature | | |
| Access | 1.408 (1.145, 1.732) [0.001] | 0.864 (0.652, 1.145) [0.310] |
| Aesthetics | 1.921 (1.594, 2.314) [<0.001] | 1.062 (0.838, 1.345) [0.619] |
| Amenities | 1.391 (1.121, 1.727) [0.003] | 0.721 (0.540, 0.962) [0.026] |
| Facilities | 1.610 (1.299, 1.997) [<0.001] | 0.926 (0.670, 1.278) [0.639] |
| Incivilities | 1.524 (1.302, 1.785) [<0.001] | 1.029 (0.825, 1.285) [0.797] |
| Potential usage | 3.631 (2.803, 4.704) [<0.001] | 3.510 (2.434, 5.062) [<0.001] |
| Safety | 1.759 (1.464, 2.114) [<0.001] | 1.402 (1.143, 1.719) [0.001] |
| Social | 1.823 (1.491, 2.230) [<0.001] | 1.242 (0.943, 1.634) [0.122] |
| Visited preferred nearby natural setting at least once a week for the last 4 weeks | | |
| Access | 2.067 (1.491, 2.864) [<0.001] | 1.235 (0.814, 1.876) [0.321] |
| Aesthetics | 2.156 (1.600, 2.904) [<0.001] | 0.974 (0.679, 1.397) [0.887] |
| Amenities | 1.730 (1.203, 2.489) [0.003] | 0.748 (0.493, 1.135) [0.172] |
| Facilities | 1.913 (1.387, 2.638) [<0.001] | 0.930 (0.601, 1.437) [0.742] |
| Incivilities | 1.499 (1.192, 1.886) [0.001] | 0.798 (0.599, 1.064) [0.125] |
| Potential usage | 5.399 (3.408, 8.553) [<0.001] | 5.521 (3.020, 10.094) [<0.001] |
| Safety | 2.255 (1.649, 3.082) [<0.001] | 1.782 (1.268, 2.506) [0.001] |
| Social | 1.988 (1.425, 2.773) [<0.001] | 1.108 (0.726, 1.693) [0.634] |
| Visited nature more often since the onset of the COVID-19 pandemic than before | | |

**Table 5.** *Cont.*

|  | Single Quality Domain Model | Multi Quality Domain Model |
| --- | --- | --- |
| Access | 1.683 (1.352, 2.095) [<0.001] | 1.148 (0.876, 1.505) [0.317] |
| Aesthetics | 1.685 (1.398, 2.031) [<0.001] | 1.039 (0.811, 1.330) [0.764] |
| Amenities | 1.599 (1.274, 2.008) [<0.001] | 0.964 (0.723, 1.285) [0.802] |
| Facilities | 1.729 (1.378, 2.171) [<0.001] | 1.043 (0.759, 1.433) [0.795] |
| Incivilities | 1.373 (1.159, 1.627) [<0.001] | 0.868 (0.699, 1.079) [0.203] |
| Potential usage | 2.821 (2.205, 3.611) [<0.001] | 2.405 (1.695, 3.412) [<0.001] |
| Safety | 1.359 (1.137, 1.624) [0.001] | 1.071 (0.886, 1.294) [0.480] |
| Social | 1.792 (1.476, 2.176) [<0.001] | 1.248 (0.964, 1.615) [0.092] |
| Nature has helped me stay connected with my neighbours during the COVID-19 pandemic | | |
| Access | 1.819 (1.492, 2.216) [<0.001] | 1.191 (0.921, 1.539) [0.182] |
| Aesthetics | 1.913 (1.599, 2.290) [<0.001] | 1.188 (0.944, 1.495) [0.142] |
| Amenities | 1.697 (1.370, 2.102) [<0.001] | 0.907 (0.694, 1.186) [0.476] |
| Facilities | 1.840 (1.486, 2.279) [<0.001] | 0.877 (0.656, 1.173) [0.377] |
| Incivilities | 1.688 (1.434, 1.986) [<0.001] | 1.084 (0.882, 1.330) [0.444] |
| Potential usage | 2.863 (2.279, 3.598) [<0.001] | 1.699 (1.227, 2.351) [0.001] |
| Safety | 1.515 (1.280, 1.793) [<0.001] | 1.147 (0.956, 1.377) [0.139] |
| Social | 2.304 (1.897, 2.798) [<0.001] | 1.670 (1.285, 2.171) [<0.001] |
| Nature has helped me to find solace during the COVID-19 pandemic | | |
| Access | 2.200 (1.804, 2.682) [<0.001] | 1.134 (0.864, 1.489) [0.365] |
| Aesthetics | 2.698 (2.258, 3.224) [<0.001] | 1.124 (0.892, 1.416) [0.322] |
| Amenities | 2.193 (1.784, 2.696) [<0.001] | 0.885 (0.667, 1.175) [0.398] |
| Facilities | 2.215 (1.804, 2.720) [<0.001] | 0.805 (0.588, 1.101) [0.174] |
| Incivilities | 2.039 (1.757, 2.366) [<0.001] | 1.112 (0.904, 1.370) [0.315] |
| Potential usage | 8.275 (6.330, 10.816) [<0.001] | 6.358 (4.494, 8.996) [<0.001] |
| Safety | 1.792 (1.519, 2.114) [<0.001] | 1.205 (0.998, 1.456) [0.053] |
| Social | 2.583 (2.137, 3.122) [<0.001] | 1.311 (1.008, 1.705) [0.044] |
| I visit nature to walk and exercise more often during the COVID-19 pandemic than before | | |
| Access | 2.001 (1.605, 2.494) [<0.001] | 1.418 (1.081, 1.860) [0.012] |
| Aesthetics | 1.805 (1.502, 2.170) [<0.001] | 1.120 (0.875, 1.432) [0.369] |
| Amenities | 1.706 (1.361, 2.137) [<0.001] | 0.952 (0.715, 1.267) [0.736] |
| Facilities | 1.819 (1.452, 2.278) [<0.001] | 0.948 (0.689, 1.304) [0.743] |
| Incivilities | 1.441 (1.218, 1.706) [<0.001] | 0.860 (0.693, 1.068) [0.172] |
| Potential usage | 2.966 (2.336, 3.765) [<0.001] | 2.163 (1.534, 3.051) [<0.001] |
| Safety | 1.427 (1.193, 1.707) [<0.001] | 1.100 (0.910, 1.330) [0.325] |
| Social | 1.980 (1.629, 2.407) [<0.001] | 1.363 (1.047, 1.774) [0.021] |

SE: Standard Error; 95%CI: 95% Confidence Interval; All models weighted for national representativeness and adjusted for gender, age group, country of birth, language spoken at home, relationship status, highest educational qualification, annual household income, economic status, perceived financial difficulty, housing status, and geographic region.

## 4. Discussion

Key findings from this study affirm our hypothesis on the importance of having higher quality green and blue spaces nearby and their positive role in enabling people to keep connected with their neighbours, feel a sense of solace and maintain or increase their level of physical activity during the COVID-19 pandemic. These results present a major advance in research on nature and various aspects of mental, physical and social health experienced during the pandemic that has largely ignored the issue of quality [28]. Our results show—as many already suspected—that simply having green and/or blue space nearby is not always sufficient to elicit favorable outcomes [38,44]. This is perhaps no more vividly illustrated than the over nine-fold increase in the odds of finding solace through nature during the pandemic where those natural settings were of the highest quality quintile versus the lowest. The dose–response patterns for each of the outcomes with respect to the overall quality of nearby nature spaces demonstrate how this is not a curvilinear effect, with higher quality scores and odds of favorable outcomes following positive linearity.

Analyzing separate domains of nature space quality revealed which ones appeared to be of greater importance to specific outcomes. This is important as a common policy-

relevant area for improvement with current research on perceived green space quality and various health outcomes is that data are often insufficient to discern which qualities matter most and least [45,46]. It was notable that the domain describing potential usage was consistently and positively associated with visitation and felt benefits. This is expected, as indicators in this domain emphasized nature as a setting for rest, recuperation, restoration of depleted cognitive resources, physical activity and as a play space for children. These indicators might attend more to some participants' intrinsic motivations for seeking time in nature in comparison to other domains where descriptive elements are present, but might not necessarily be of fundamental importance to the individual responding. This domain and others that were associated with some outcomes but not all, such as the safety and social domains, appeared to be more important than others, such as incivilities, facilities and aesthetics that were no longer statistically significant in multi-domain models. To some extent, this will be due to partial overlap between each of the domains, most notably for the aesthetics and potential usage domains; the presence of wildlife and other cues that prompt interest and exploration in the aesthetic domain are concomitant with the desire to feel away from the day-to-day routine of cognitive demands.

Our survey included a partial assessment of biodiversity through a single indicator. This is important as there are now several studies indicating how objective and subjectively-measured biodiversity of green spaces (e.g., presence of birds and species diversity indicators) may be an important conduit by which some mental health benefit occurs [47–50], not least through attending to people's preferences [51]. However, the present indicator likely only grazes the surface of this concept and this is an area in need of further research. Similarly, it is also plausible that some of the domains, such as incivilities, might be underpowered, given its current single-item focus on quality and maintenance, wherein specific issues might resonate strongly with some people (e.g., the presence of dog feces on grass). So too might be the access domain, with its focus more on getting to the natural space, rather than the accessibility within it, which might be especially important for people with physical limitations or disabilities [52–54]. Curiously, the amenities domain was statistically significantly and negatively associated with time in nature and non-significantly negatively associated with all other outcomes after adjusting for other domains. Caution is needed in interpreting this result. On one hand, it may be driven by multicollinearity, but on the other, negative association may be due to some people actively avoiding natural settings that are within close proximity to retail strips and similar that may attenuate their restorative experiences while in nature. For instance, while some work indicates that non-natural sounds emanating from automobiles, trucks and other elements concomitant with commercial (and industrial) landscapes may be soothed by being in nature via psychoacoustic pathways [55,56], it is unclear if such annoyances are ameliorated entirely. Furthermore, other physical cues that are either located nearby or encroach within natural settings, such as neon signage to advertise workplaces and fast-food restaurants, may also be a source of distraction (perhaps, even irritation) for many people when visiting nature for rest and escape from the day-to-day demands in life.

A third key finding was that the degree of the socioeconomic gradient in the availability of quality nature spaces was atypical, with total quality scores being only slightly higher on average for people with university-level qualifications or annual household incomes over AUD 150 k. These differences were not statistically significant in fully adjusted models. However, importantly, there was a somewhat greater gap in mean total quality scores between people whose financial situation was difficult, in comparison with those whose situation was more comfortable. These differences remained statistically significant after adjustment. How a person feels about their financial situation is important and often overlooked in epidemiological studies of person-level data that tend to rely on education, income, and in some contexts, occupational class [57,58]. While difficult financial circumstances were more common in participants with less than 12 years of education versus those with a bachelor's degree (9.9% vs. 6.7%) or among those with AUD 50 k per annum or less versus those with AUD 151 k per annum or greater (17.8% vs. 2.63%),

clearly there are situations in which those with higher qualifications and income categories may also be living under major financial strain and vice versa. This variable, therefore, provides more incisive utility for accounting for socioeconomic circumstances than other more routinely-used variables. However, it is also worth noting that how people perceive the quality of their nearby green and blue spaces may be, in part, influenced by the levels of financial and psychological strain under which they are presently living. Accordingly, further work might examine to what extent changes in perceived financial circumstances among people whose socioeconomic circumstances remain consistent, as measured by income and education (etc.), may influence how they perceive the natural spaces they have nearby.

It is worthwhile noting that although our survey was able to measure time spent in nature, which is important as previous studies have indicated [59,60], we did not have information on what survey participants did in those spaces specifically beyond the outcomes already analyzed. For instance, some people may have visited specific parks on Saturdays to engage with other local community members in the Parkrun movement, which various studies indicate provides opportunities for volunteering and social connection, as well as physical activity [61–63]. For many people, especially during the COVID-19 pandemic, cemeteries may have played a key role in giving people opportunities to be outdoors and connect, if not with each other, then with loved ones no longer around [64]. Others may have visited natural settings they regard as special places, perhaps due to childhood memories [65,66] or as providing opportunities to do things they feel unable to at home, such as connecting with peers through allotment gardens [67,68]. Further qualitative research and maybe further survey analysis are needed to better understand the roles in which specific types of green space and their qualities have aided coping and restoration through the pandemic.

An additional layer to this research is the well-reported socioeconomic inequities in green space and blue space availability and qualities are likely to reflect, in part, personal preferences, financial capacities and willingness to pay to live near these health-promoting resources. The intersection between these economic issues and the epidemiological literature remains a gap in knowledge, though research was carried out on these aspects. For example, Johnson and Thomassin [69] provided a model to estimate the willingness to pay for surface water quality improvements by recreational users. Further investigation utilizing longitudinal data capable of tracking changes in the qualities of green and blue spaces that occur and impacts on both the health of local residents and on population flows in and out of the areas nearby is warranted.

Beyond the strengths and limitations already discussed, this study benefits from a large and nationally representative sample of the Australian adult population, covering all states and territories. The survey contained a large range of variables used to describe green and blue spaces, permitting the identification of nature space qualities and the development of an overall score from domains established by a published study that focused on in-person auditing of parks [42]. All descriptive statistics and models were adjusted using a comprehensive set of confounding variables and also weighted to ensure parameter estimates that can be extrapolated to the adult population of Australia. The analyses use data of cross-sectional design and so the associations reported should not be interpreted as definitively revealing cause and effect. Follow-up of the same individuals over time will enable stronger epidemiological study designs with which to minimize the potential for reverse causation, wherein individuals more prone to poorer health are socioeconomically disadvantaged in part because of their circumstances and move into areas with poorer quality green and blue spaces as a result. These data and follow-up of the same individuals will also permit opportunities to study the longer-term impacts of the COVID-19 pandemic and to identify the extent to which different types of nature, preferential elements of it and ways in which people interact with it have supported recovery and flourishing.

**Author Contributions:** Conceptualization, X.F. and T.A.-B.; methodology, X.F. and T.A.-B.; software, T.A.-B.; validation, T.A.-B.; formal analysis, X.F. and T.A.-B.; investigation, X.F. and T.A.-B.; resources,



X.F. and T.A.-B.; data curation, X.F. and T.A.-B.; writing—original draft preparation, X.F. and T.A.-B.; writing—review and editing, X.F. and T.A.-B.; visualization, X.F. and T.A.-B.; project administration, X.F. and T.A.-B.; funding acquisition, X.F. and T.A.-B. All authors have read and agreed to the published version of the manuscript.

**Funding:** This work was supported by the Hort Frontiers Green Cities Fund, part of the Hort Frontiers strategic partnership initiative developed by Hort Innovation, with co-investment from the University of Wollongong (UOW) Faculty of Arts, Social Sciences and Humanities, the UOW Global Challenges initiative and contributions from the Australian Government (project number #GC15005). T.A.-B. was supported by a National Health and Medical Research Council Boosting Dementia Research Leader Fellowship (#1140317). X.F. was supported by a National Health and Medical Research Council Career Development Fellowship (#1148792). All aspects related to the conduct of this study including the views stated and the decision to publish the findings are that of the authors only.

**Institutional Review Board Statement:** The study was conducted according to the guidelines of the Declaration of Helsinki, and approved by the Ethics Committee of the University of Wollongong (protocol code 2020/343, 14 September 2020).

**Informed Consent Statement:** Informed consent was obtained from all subjects involved in the study.

**Data Availability Statement:** The data are not publically available.

**Acknowledgments:** We thank the Social Research Centre and the Life in AustraliaTM panel members. We thank Richard Mitchell (University of Glasgow) for sharing ideas on survey questions.

**Conflicts of Interest:** The authors declare no conflict of interest. The funders had no role in the design of the study; in the collection, analyses, or interpretation of data; in the writing of the manuscript, or in the decision to publish the results.

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
