# Peer review of "Perceived Qualities, Visitation and Felt Benefits of Preferred Nature Spaces during the COVID-19 Pandemic in Australia: A Nationally-Representative Cross-Sectional Study of 2940 Adults"

_land, doi:10.3390/land11060904_

Round 1

Author Response

R3

Very clear and well discussed and organised paper and very well developed all through. I am convinced by the specific claims of the paper that issues of green/blue space quality not as fully developed as it could be, so good to see this specific dimension explored fully and core of paper built around this

AUTHORS: Thank you for your positive and constructive review of our manuscript.

Sample very clear and queries on geography well identified and provenance of sample cohort and its generally sound representativeness well discussed.

AUTHORS: Thank you

Specific questions that respondents answered were well organised and an intriguing range of themes identified across sections 2.2 and 2.3 and listed in Table 1. The only thing I might ask for would be a little more rationale around some of choices (maybe backed up by some literature, unless I missed this). I am happy with them and they are some very interesting questions in there, such as incivilities etc.

AUTHORS: We have added a line to explain that where possible, some statements were adopted or derived from existing literature. We provide an example from the aesthetics domain and also add a citation to Hartig’s perceived restorativeness scale. This is as follows:

“Where possible, these questions were derived from existing literature. For instance, “there is much to explore and discover there” in the aesthetics domain was drawn from Hartig’s perceived restorativeness scale [43].”

One last query might be the specific phrasing of the questions, which do appear to be positive statements. I assume there is a logic to keeping to one line of questioning, i.e. not confusing a positive statement with a negative one, but could just clarify that in the text.

AUTHORS: This is correct; positive framing was consistent to enable survey respondents to quickly answer. Our consultation with expert survey designers indicated that mixing positive and negative framing can confuse survey respondents and increase the risk of non-response or drop-out.

Results quite clear across the paper with summary stats well presented in Table 4 and Figure 1 in Section 3.2 especially strong in relation to the visual presentation and consistency of results in relation to the quintile scores and nice to see visual identification of Cis as well. One possibly addition to explain to an non-expert might be the specific meaning of a response with long as opposed to short bars, though the odds ratios speak for themselves. In Table 5 the high scores for potential usage and wondered if the authors could just add a few sentences more on why this might be?

AUTHORS: We have added wording intended to aid interpretation of the solace-focussed estimate that had an odds ratio of approximately 9 and a wide confidence interval. This new text is:

“In order of magnitude, participants in quintile 5 compared with quintile 1 for nature space total quality scores had 9.5 times greater odds of reporting that time in nature had been a source of solace during the pandemic (OR 9.49, 95%CI 6.73-13.39; wider confidence intervals are in part indicative of smaller sample sizes)…”

In terms of potential usage results in table 5, we have added wording in the results to clarify that the odds ratios were also consistently stronger than other domains. This text is below.

The magnitude of odds ratios for potential usage were also consistently larger than other domains.

We also add a sentence to speculate on why potential usage might be stronger than other domains in the Discussion section. This text is below.

“These indicators might attend more to some participants’ intrinsic motivations for seeking time in nature in comparison to other domains that descriptive elements that are present, but might not necessarily be of fundamental importance to the individual responding.”  

Discussion section very clear and liked the statements of core findings but also a recognition of some of the limitations of the work and suggestions as to ways in which a deeper qualitative nuance might be developed and followed up on.

AUTHORS: Thank you.

Reviewer 2 Report

Overall, I think the merit of t his study is very high. Economists have tried to assess how people value nature - this is not new - but they have not broken the valuation down into specific components of why people might value natural spaces. However, the authors to do not acknowledge this practice in economics, which both supports their contention that the quality of nature is not considered in previous studies in the detail that they provide, but it is considered in a more aggregate manner in these studies. I would like to see an additional paragraph acknowledging and citing some of this work. Johnson and Thomassin (2010) provide widely used model to estimate the willingness-to-pay for water quality by recreational users. It is based on a meta-analysis of many North American studies. Although the model is probably out of date, it is still widely used by economists in North America. Johnston, R.J., Thomassin, P.J., 2010. Willingness to pay for water quality improvements in the United States and Canada: Considering possibilities for international meta-analysis and benefit transfer. Agricultural and Resource Economic Review, 39(1), 114-131.

The authors group the responses from the subjects using  Likert scale, and then treat the scale as interval data and further treat the data. Although this is incorrect, many studies have demonstrated that using parametric tests does not alter the results of the analysis. However, the most recent discussions on this issue also show that there are no reasons not to use the nonparametric statistics. I think that it is important for the authors to explain their use of parametric statistics.

Table 4 should be included as an appendix. In the body of the article, the authors should use a bar graph, using only their reference categories from Table 4

Johnston, R.J., Thomassin, P.J., 2010. Willingness to pay for water quality improvements in the United States and Canada: Considering possibilities for international meta-analysis and benefit transfer. Agricultural and Resource Economic Review, 39(1), 114-131.

Author Response

R1

Overall, I think the merit of t his study is very high. Economists have tried to assess how people value nature - this is not new - but they have not broken the valuation down into specific components of why people might value natural spaces. However, the authors to do not acknowledge this practice in economics, which both supports their contention that the quality of nature is not considered in previous studies in the detail that they provide, but it is considered in a more aggregate manner in these studies. I would like to see an additional paragraph acknowledging and citing some of this work. Johnson and Thomassin (2010) provide widely used model to estimate the willingness-to-pay for water quality by recreational users. It is based on a meta-analysis of many North American studies. Although the model is probably out of date, it is still widely used by economists in North America. Johnston, R.J., Thomassin, P.J., 2010. Willingness to pay for water quality improvements in the United States and Canada: Considering possibilities for international meta-analysis and benefit transfer. Agricultural and Resource Economic Review, 39(1), 114-131.

AUTHORS: Thank you for your thoughtful and constructive review. We were not privy to this particular contribution in economics and have followed your recommendation to cite it in a new paragraph in the Discussion section, as follows:

“An additional layer to this research is the well-reported socioeconomic inequities in green space and blue space availability and qualities are likely to reflect, in part, personal preferences, financial capacities and willingness-to-pay to live near these health-promoting resources. The intersection between these economic issues and the epidemiological literature remains a gap in knowledge, though research has been done on some aspects. For example, Johnson and Thomassin [68] provided a model to estimate the willingness-to-pay for surface water quality improvements by recreational users. Further investigation utilizing longitudinal data capable of tracking changes in the qualities of green and blue spaces that occur and impacts on both the health of local residents and on population flows in and out of the areas nearby is warranted.”

The authors group the responses from the subjects using  Likert scale, and then treat the scale as interval data and further treat the data. Although this is incorrect, many studies have demonstrated that using parametric tests does not alter the results of the analysis. However, the most recent discussions on this issue also show that there are no reasons not to use the nonparametric statistics. I think that it is important for the authors to explain their use of parametric statistics.

AUTHORS: As the reviewer highlights, many studies demonstrate the results of parametric or non-parametric tests are similar. Accordingly, the relatively simpler parametric test was used.

Table 4 should be included as an appendix. In the body of the article, the authors should use a bar graph, using only their reference categories from Table 4

AUTHORS: We appreciate the suggestion, though as the contents of Table 4 are describing the distribution of the nature quality variable across sample characteristics, we think this is essential to retain in the main body of the article. We retain the table, which is a more concise way or reporting the data, rather than producing a large number of bar graphs.

Johnston, R.J., Thomassin, P.J., 2010. Willingness to pay for water quality improvements in the United States and Canada: Considering possibilities for international meta-analysis and benefit transfer. Agricultural and Resource Economic Review, 39(1), 114-131.

AUTHORS: We have now cited this paper.

Reviewer 3 Report

The authors aimed to investigate how the quality of green and blue space influences the level of their visitation during the covid19 pandemic in Australia. The investigation was performed through telephone and on line survey and included responses of  about 3000 adult residents. Authors used Likert scale for classification of responses, which were grouped into eight quality domains such as: access, aesthetics, amenities, facilities, incivilities, potential usage, safety and social. These domains were then summed into a nature quality score. Relations between the quality variables and a range of nature visitation and benefits were tested using weighted models, adjusted for demographic and socioeconomic confounders.

Generally the responders claimed increased levels of nature visitation during  the COVID-19 pandemic than before,  and their visits to  nature enabled them to take solace and respite.

Authors concluded basing on the  weighted logistic regression results, time  spent in nature spaces and frequency of visiting them were the most affected by potential usage and safety. The survey performed by Authors  contained a large range of variables used to describe green and blue spaces, permitting the identification of nature space qualities and the development of an overall score from established domains. However, Authors were aware of the fact that the analyses used data of cross-sectional design and the associations reported should not be interpreted as definitively revealing cause and effect. Moreover they have not investigated what survey participants did in the green or blue spaces they visited.

The paper is generally interesting and well written. It contains some new approaches to well- known positive effect of green and blue spaces on human health and particularly mental health. The hypothesis is not formulated, so I think it should be added and referred in the discussion section. In my opinion when evaluating quality of green and blue spaces a little stronger focus should have been put on the nature and their biodiversity especially in the aesthetic or incivilities domains, in which I found the statements to consider by the responders very general.  In multi quality domain model  both aesthetics and incivilities resulted statistically  unimportant which seems strange  when evaluating the quality especially of a green space.

Author Response

R2

The authors aimed to investigate how the quality of green and blue space influences the level of their visitation during the covid19 pandemic in Australia. The investigation was performed through telephone and on line survey and included responses of  about 3000 adult residents. Authors used Likert scale for classification of responses, which were grouped into eight quality domains such as: access, aesthetics, amenities, facilities, incivilities, potential usage, safety and social. These domains were then summed into a nature quality score. Relations between the quality variables and a range of nature visitation and benefits were tested using weighted models, adjusted for demographic and socioeconomic confounders. Generally the responders claimed increased levels of nature visitation during  the COVID-19 pandemic than before,  and their visits to  nature enabled them to take solace and respite. Authors concluded basing on the  weighted logistic regression results, time  spent in nature spaces and frequency of visiting them were the most affected by potential usage and safety. The survey performed by Authors  contained a large range of variables used to describe green and blue spaces, permitting the identification of nature space qualities and the development of an overall score from established domains. However, Authors were aware of the fact that the analyses used data of cross-sectional design and the associations reported should not be interpreted as definitively revealing cause and effect. Moreover they have not investigated what survey participants did in the green or blue spaces they visited.

AUTHORS: Thank you for your thoughtful and constructive review. This point on what people did while visiting green and blue spaces is covered in a paragraph in the Discussion section:

“It is worthwhile noting that although our survey was able to measure time spent in nature, which is important as previous studies have indicated [58, 59], we did not have information on what survey participants did in those spaces specifically beyond the outcomes already analyzed. For instance, some people may have visited specific parks on Saturdays to engage with other local community members in the Parkrun movement, which various studies indicate provides opportunities for volunteering and social connection, as well as physical activity [60-62]. For many people, especially during the COVID-19 pandemic, cemeteries may have played a key role in giving people opportunities to be outdoors and connect, if not with each other, then with loved ones no longer around [63]. Others may have visited natural settings they regard as special places, perhaps due to childhood memories [64, 65] or as providing opportunities to do things they feel unable to at home, such as connecting with peers through allotment gardens [66, 67]. Further qualitative research and maybe further survey analysis is needed to better understand the roles in which specific types of green space and their qualities have aided coping and restoration through the pandemic.”

The paper is generally interesting and well written. It contains some new approaches to well- known positive effect of green and blue spaces on human health and particularly mental health. The hypothesis is not formulated, so I think it should be added and referred in the discussion section. In my opinion when evaluating quality of green and blue spaces a little stronger focus should have been put on the nature and their biodiversity especially in the aesthetic or incivilities domains, in which I found the statements to consider by the responders very general.  In multi quality domain model  both aesthetics and incivilities resulted statistically  unimportant which seems strange  when evaluating the quality especially of a green space.

AUTHORS: Thank you. Several of these points are addressed in a paragraph in the Discussion section, where we reflect on the need for future work to have more detailed coverage of issues related to biodiversity, aesthetics, incivilities, etc. This paragraph is copied in below. We have also added a hypothesis as the last line of the Introduction as follows:

“We hypothesized that people tended to visit and benefit more from green and/or blue spaces that they felt had more of the qualities they value.”

Paragraph in the Discussion section:

“Our survey included partial assessment of biodiversity through a single indicator. This is important as there are now several studies indicating how objective and subjectively-measured biodiversity of green spaces (e.g. presence of birds and species diversity indicators) may be an important conduit by which some mental health benefit occurs [46-49], not least through attending to people’s preferences [50]. However, the present indicator likely only grazes the surface of this concept and this is an area in need of further research. Similarly, it is also plausible that some of the domains, such as incivilities, might be underpowered, given its current single-item focus on quality and maintenance, wherein specific issues might resonate strongly with some people (e.g. the presence of dog feces on grass). So too might be the access domain, with its focus more on getting to the nature space, rather than the accessibility within it, that might be especially important for people with physical limitations or disabilities [51-53]. Curiously, the amenities domain was statistically significantly and negatively associated with time in nature and non-significantly negatively associated with all other outcomes after adjusting for other domains. Caution is needed in interpreting this result. On one hand it may be driven by multicolinearity, but on the other, negative association may be due to some people actively avoiding natural settings that are within close proximity to retail strips and similar that may attenuate their restorative experiences while in nature. For instance, while some work indicates that non-natural sounds emanating from automobiles, trucks and other elements concomitant with commercial (and industrial) landscapes may be soothed by being in nature via psychoacoustic pathways [54, 55], but it is unclear if such annoyances are ameliorated entirely. Furthermore, other physical cues that are either located nearby or encroach within natural settings, such as neon signage to advertise workplaces and fast food restaurants, may also be a source of distraction (perhaps, even irritation) for many people when visiting nature for rest and escape from the day-to-day demands in life.”

This manuscript is a resubmission of an earlier submission. The following is a list of the peer review reports and author responses from that submission.